# A Long Noncoding RNA Derived from lncRNA–mRNA Networks Modulates Seed Vigor

**DOI:** 10.3390/ijms23169472

**Published:** 2022-08-22

**Authors:** Qiaoli Gao, Jinzhao Liu, Huibin Weng, Xi Yuan, Wuming Xiao, Hui Wang

**Affiliations:** National Engineering Research Center of Plant Space Breeding, South China Agricultural University, Guangzhou 510642, China

**Keywords:** rice, lncRNA, seed vigor, low temperature, SAUR

## Abstract

The discovery of long noncoding RNAs (lncRNAs) has filled a great gap in our understanding of posttranscriptional gene regulation in a variety of biological processes related to plant stress responses. However, systematic analyses of the lncRNAs expressed in rice seeds that germinate under cold stress have been elusive. In this study, we performed strand-specific whole transcriptome sequencing in germinated rice seeds under cold stress and normal temperature. A total of 6258 putative lncRNAs were identified and expressed in a stage-specific manner compared to mRNA. By investigating the targets of differentially expressed (DE) lncRNAs of LT-I (phase I of low temperature)/NT-I (phase I of normal temperature), it was shown that the auxin-activated signaling pathway was significantly enriched, and twenty-three protein-coding genes with most of the members of the SAUR family located in chromosome 9 were identified as the candidate target genes that may interact with five lncRNAs. A seed vigor-related lncRNA, SVR, which interplays with the members of the SAUR gene family in cis was eventually identified. The CRISPR/Cas 9 engineered mutations in SVR cause delay of germination. The findings provided new insights into the connection between lncRNAs and the auxin-activated signaling pathway in the regulation of rice seed vigor.

## 1. Introduction

Rice (*Oryza sativa* L.) is the main food crop worldwide. The planting cost of rice is severely affected by the cultivation method, and the approach of direct seeding has a lower cost and has become prevalent worldwide [1]. Seed vigor is a complex and important agricultural trait that determines the potential for rapid, uniform emergence and the establishment of strong seedlings in any environmental condition [2]. Strong seed vigor under low-temperature conditions is important in direct sowing rice production systems and particularly important for rice production in temperate rice-growing areas, and in areas with a cold irrigation water supply [3].

Eukaryotic genomes are not simple and well-ordered substrates of gene transcription [4]. Transcriptome studies have shown that >90% of the genomes are transcribed, and a billion transcripts correspond to noncoding (nc) RNAs [5]. The widespread occurrence of ncRNAs suggests that they are functionally important. Studying ncRNAs facilitates enhancing the understanding of genome organization and regulation [6]. Noncoding RNAs can be classified into housekeeping ncRNAs and regulatory ncRNAs. Housekeeping ncRNAs contain small nuclear, small nucleolar, and transfer RNAs. Regulatory ncRNAs include short regulatory ncRNAs and long regulatory ncRNAs. Short regulatory ncRNAs include microRNAs, small interfering RNAs, and Piwi-associated RNAs [7]. The other type of ncRNA is long noncoding RNAs (lncRNAs) [4]. LncRNAs are defined as RNAs longer than 200 nucleotides holding little or no protein-coding capacity [8]. LncRNAs can be classified into five categories according to their location relative to protein-coding genes: (1) sense, (2) antisense, (3) bidirectional, (4) intronic, and (5) intergenic [4,9]. Most lncRNAs are transcribed by RNA polymerase Ⅱ as mRNA, have 7-methyl guanosine caps at the 5′-end and 3′-end poly (A) tails, and are presumed to be processed similarly to mRNAs [10]. There are several pathways that lncRNAs may originate from: (A) a protein-coding gene frame is disrupted and then transformed into a ncRNA; (B) by chromosome arrangement, where two separated and untranscribed regions are joined together and form a ncRNA that contains multiple exons; (C) retrotransposition is replicated and forms functional ncRNAs or pseudogenes without functions; (D) ncRNAs contain adjacent repeats that are generated from tandem repeats; and (E) the insertion of a transposable element gives rise to a functional ncRNA [4,11]. LncRNAs can interact with the adjacent protein-coding genes in cis or interact with the distant protein-coding genes in trans action [10]. Previous studies have shown that lncRNAs can regulate gene expression at the level of chromatin modification, transcription, and post-transcriptional processing [10].

Although not much progress has been made in the study of plant lncRNAs, we cannot ignore the roles they play in plants. The lncRNA *MIS-SHAPEN ENDOSPERM (MISSEN)* is a maternally expressed cytoplasm-localized lncRNA. *MISSEN* suppresses endosperm development by negatively regulating the function of the helicase family protein (HeFP), leading to a prominent dent and bulge in the seed. Its expression is controlled by histone H3 lysine 27 trimethylation (H3K27me3) modification [12]. *LEUCINE-RICH REPEAT RECEPTOR KINASE ANTISENSE INTERGENIC RNA (LAIR)* is a rice grain yield-related lncRNA that is transcribed from the antisense strand of the neighboring *LRK* (*leucine-rich repeat receptor kinase*) gene cluster. Overexpression of *LAIR* can increase grain yield and upregulate the expression of several *LRK* via epigenetic modifications [13]. *LONG-DAY-SPECIFIC MALE-FERTILITY-ASSOCIATED RNA* (*LDMAR*) is a long-day-specific male-fertility-associated lncRNA that is 1236 bases in length, and regulates the photoperiod-sensitive male sterility (PSMS) of rice. A single nucleotide polymorphism (SNP) alters the secondary structure of *LDMAR* that may bring about increased methylation in the promoter region of *LDMAR* and reduce the transcription of *LDMAR* specifically under long-day conditions, resulting in premature programmed cell death in developing anthers, thus causing PSMS [14]. Another PSMS-related lncRNA is *PHOTOPERIOD-SENSITIVE GENIC MALE STERILITY 1* (*PMS1T*), which is targeted by miR2118 to produce 21-nt phasiRNAs that preferentially accumulate in the PSMS line under long-day conditions [15]. In Arabidopsis, *FLOWERING LOCUS C (FLC)* functions in the vernalization pathway. *FLC* represses the expression of floral integrators such as *FT*, *FD*, and *SOC1* required for flowering and negatively regulates flowering [16]. The regulation of flowering time was mainly via the modification of histones [17,18,19]. The lncRNAs *COLD ASSISTED INTRONIC NONCODING RNA (COLDAIR)* and *COLD INDUCED LONG ANTISENSE INTRAGENIC RNA (COOLAIR)* are transcribed from *FLC* and function in *FLC* via epigenetic silencing [20,21]. *COOLAIR* is transcribed from the 3′ end of *FLC* in the antisense direction, which is alternatively spliced and polyadenylated [22], and functions in coordinated switching of chromatin states at *FLC* that occurs during cold, leading to transcriptional shutdown with epigenetic silencing [20]. *COLDAIR* is an intronic ncRNA that is transcribed from the sense direction of *FLC* and contains a 5′ cap structure, but is not polyadenylated [21]. In *COLDAIR* knockdown lines, cold-mediated H3K27me3 enrichment is largely impaired. During vernalization, *COLDAIR* mainly works in the recruitment of PRC2 to *FLC* chromatin to establish the stable silencing of *FLC* [21].

Auxins play a critical role in most major growth responses in plants and control diverse aspects of plant growth [23]. Auxins can play roles in cell division, extension, and differentiation in part by altering gene expression [24]. Rapidly mediating changes in cell expansion is one of the most striking effects of auxin [23]. Many genes are specifically and rapidly induced following auxin application—these genes are referred to as early auxin response genes, and can fall into three major gene families: the *auxin*/*indoleacetic acid* (*Aux*/*IAA*), *Gretchen Hagen-3* (*GH3*), and *small auxin-up RNA* (*SAUR*) gene families [24]. SAURs were originally identified in elongating soybean hypocotyl sections [25]. *SAURs* are the largest family of early auxin response genes—there are 81 *SAUR* (including two pseudogenes) genes in *Arabidopsis* [24], 58 *SAUR* genes (including two pseudogenes) in rice [26], and 79 *SAUR* genes in maize [27]. *OsSAUR39* and *OsSAUR45* were identified to affect auxin synthesis and transport in rice [28,29]. *OsSAUR33* was proven to affect rice seed vigor through the sugar pathway [30]. It has long been hypothesized that SAURs may be involved in auxin-regulated cell expansion [31] and the overexpression of *SAUR36* [32], *SAUR41* [33], and the stabilized fusion proteins *SAUR19* [34] and *SAUR63*, promoting hypocotyl elongation as a result of increased cell expansion in *Arabidopsis* [35]. The mechanism underlying SAUR-mediated cell expansion was proposed by Spartz et al. [36], who suggested that SAURs promote cell expansion via an acid growth mechanism, and that these proteins inhibit PP2C.D family phosphorylation to activate plasma membrane (PM) H^+^-ATPases and thereby promote cell expansion.

In this study, rice seeds were germinated under normal temperature (25 °C, NT) and low temperature (15 °C, LT), and they were collected as samples for strand-specific whole-transcriptome sequencing. We aimed to determine whether lncRNAs played roles in seed vigor and how they worked. The findings reveal the involvement of lncRNAs in seed vigor under LT and provide useful information in the further molecular function analysis of lncRNAs.

## 2. Results

### 2.1. RNA Sequencing and Identification of lncRNAs

To systematically identify lncRNAs related to rice seed vigor, we constructed 15 rRNA-free libraries with three replicates (Appendix A). The samples were obtained from rice seeds germinated under LT (15 °C) and NT (25 °C) conditions and dry seeds. The dry seeds were designated as 0 h (Figure 1A). At the first stage, the embryo had just broken out of the glume (designated as NT-I and LT-I) (Figure 1B). The second stage was when the length of the embryo reached about 1.0 mm (designated NT-II and LT-II) (Figure 1C). Three samples at each stage of NT, LT, and 0 h were collected as three replicates, with each replicate containing 30 uniform seeds. Strand-specific paired-end deep sequencing of these 15 libraries was performed. After filtering out low-quality reads, 269.65 Gb of clean data were obtained, with an average of 16.09 Gb per sample. Then, clean data for each sample were mapped to the Japonica reference genome IRGSP-1.0 (https://rapdb.dna.affrc.go.jp/download/history/irgsp1_2019-06-26.html (accessed on 15 July 2019)) using the HISAT2 program (version 2.0.4; [37]) with an average alignment efficiency of 94.4% and Q30 higher than 95.19% (Appendix A).

A strict pipeline was used to identify the lncRNAs (Figure 2). Firstly, to obtain basic transcripts, four basic filtering steps were conducted: screening the transcripts with class codes “i”, “x”, “u”, “o” and “e”, selecting transcripts with ≥2 exons, filtering out transcripts shorter than 200 nucleotides, and selecting the transcripts with an FPKM higher than 0.1. Then, the transcripts possessing coding potential, which was evaluated using the Coding Non-Coding Index (CNCI), the Coding Potential Calculator (CPC), the Coding Potential Assessment Tool (CPAT), and Pfam-scan (version 1.3, E-value < 0.01), were removed. Finally, 6258 highly reliable putative lncRNAs were identified. A circos plot (Figure 3E) showed the locations and expression of all 6258 lncRNAs in the assembled scaffolds, which demonstrated the even distribution of the lncRNAs among the 12 chromosomes. The putative lncRNAs were further classified into four types, lincRNA, antisense-lncRNA, intronic-lncRNA, and sense-lncRNA, based on their location relative to the protein-coding gene (Figure 3F). The majority of lncRNAs were intergenic lncRNAs (lincRNAs) (4520, 72.2%), followed by sense lncRNAs (933, 14.9%) and antisense lncRNAs (638, 10.2%). Only 2.7% (167) were intronic lncRNAs. Therefore, lincRNAs, as the dominant type of lncRNAs, were the focus of this study.

### 2.2. Structural Characterization of lncRNAs

To characterize the genomic features of lncRNAs, we compared them with the mRNAs assembled in this research. The results are shown in Figure 3. The majority of lncRNAs were shorter than mRNAs in length (Figure 3A), with 78.55% of lncRNAs ranging from 400 nt to 1000 nt. In contrast, nearly all mRNAs were ≥200 nt, with 43.57% of mRNAs ranging from 400 nt to 1000 nt, 37.13% of mRNAs ranging from 600 nt to 3000 nt, and 19.30% of mRNAs > 3000 nt. The exon number of the lncRNAs ranged from two to ten, and approximately 80% of lncRNAs had only two exons. However, the exon number of the mRNAs ranged from one to more than thirty, and only 37.87% of mRNAs had just one exon. In total, the exon number of lncRNAs was less than that of mRNAs (Figure 3B). The ORF length of lncRNAs ranged from 50 nt to 500 nt, but that of mRNAs ranged from 100 nt to more than 2000 nt. Additionally, the predicted maximum ORF length among the lncRNAs was 500 bp, but the predicted maximum ORF length among the mRNAs was more than 2000 bp (Figure 3C). According to FPKM, the expression level of lncRNAs was slightly lower than that of mRNAs (Figure 3D). Thus, the results indicate that lncRNAs differed from mRNAs in terms of structure and expression level. LncRNAs were shorter than mRNAs in length, possessed fewer exons and shorter ORFs, and showed lower expression levels than mRNAs.

To evaluate the conservation of the identified lncRNAs, the lncRNAs were aligned with the known plant lncRNAs in the NONCODE database (http://www.noncode.org/index.php (accessed on 30 December 2021)) by BLASTN. Only 81 lncRNAs were comparable to the known plant lncRNAs (Appendix A), with 49 in O. *rufipogon*, 8 in cucumber, 6 in cassava, 3 in quinoa, 2 in B. *napus*, 2 in grape, 2 in maize, 2 in O. *sativa*, 2 in tomato, 1 in A. *thaliana*, 1 in B. *rapa*, 1 in banana, 1 in *p*. *trichocarpa*, and 1 in trefoil. These result indicated the low conservation of lncRNAs between other plants and Francis rice.

### 2.3. More lncRNAs Than mRNAs Are Specifically Expressed at Different Stages

The lncRNAs and mRNAs expressed in NT, LT, and 0 h are shown in Venn diagrams in Figure 4A,B. There were 164 (13.72%) lncRNAs expressed universally in the NT treatment (NT-I, NT-II), LT treatment (LT-I, LT-II), and the 0 h time point. However, 715 (59.83%), 41 (3.43%), and 76 (6.36%) lncRNAs were specifically expressed at the 0 h time point, NT and LT treatments, respectively (Figure 4A). For mRNAs, a total of 12,514 (70.77%) were commonly expressed among the NT treatment (NT-I, NT-II), LT treatment (LT-I, LT-II), and the 0 h time point. Only 1036 (5.86%) mRNAs were specifically expressed at the 0 h time point, and 428 (2.42%) and 632 (3.57%) mRNAs were specifically expressed in the NT and LT treatments, respectively (Figure 4B). The proportion of specifically expressed lncRNAs was significantly higher than that of specifically expressed mRNAs at the 0 h time point.

Further analysis was performed to compare the expressed lncRNAs and mRNAs at different stages of germination. The Venn diagrams in Figure 4C–F indicate the expressed lncRNAs and mRNAs in different stages of NT and LT. Under NT and LT treatments, the proportion of total expressed lncRNAs decreased from the 0 h time point to stage II, but the proportion of total expressed mRNAs increased slightly from the 0 h time point to stage II (Figure 4H).

Intriguingly, the proportion of specifically expressed lncRNAs was higher than that of specifically expressed mRNAs at every stage of the same sample under NT and LT treatments (Figure 4G). Under NT treatment, the number of specifically expressed lncRNAs reached 649 (46.69%), 166 (11.94%), 105 (7.55%) at 0 h, NT-I, and NT-II, respectively (Figure 4G). However, the number of specifically expressed mRNAs was 735 (3.89%), 573 (3.04%), and 1250 (6.62%) at 0 h, NT-I, and NT-II, respectively (Figure 4G). Under LT treatment, 535 (35.20%), 247 (16.25%), and 119 (7.83%) lncRNAs were expressed specifically at 0 h, LT-I, and LT-II, respectively (Figure 4G). In contrast, 749 (4.03%), 454 (2.44%), and 861 (4.64%) mRNAs were expressed specifically at 0 h, LT-I, and LT-II, respectively (Figure 4G). According to the above results, it was speculated that lncRNAs may play a unique role in seed germination.

### 2.4. Analysis of Differentially Expressed lncRNAs

Differentially expressed (DE) lncRNAs and DE mRNAs were identified by comparing the normalized expression of transcripts between different stages under NT and LT treatments (i.e., NT-I/0 h, NT-II/NT-I, LT-I/0 h, LT-II/LT-I, *p* < 0.05, |log2 (fold change)| ≥ 1). There were a total of 12,529 (NT-I/0 h, NT-II/NT-I) and 10,348 (LT-I/0 h, LT-II/LT-I) DE transcripts under NT and LT, respectively. Among them, there were 330 DE lncRNAs and 12,199 DE mRNAs under NT, while there were 199 DE lncRNAs and 10,149 DE mRNAs under LT (Figure 5A). The overwhelming majority of DE mRNAs and DE lncRNAs were observed in NT (LT)-I/0 h, including 8719 (71.47%) DE mRNAs and 289 (87.58%) DE lncRNAs in NT-I/0 h and 8999 (88.67%) DE mRNAs and 186 (93.47%) DE lncRNAs in LT-I/0 h. The NT-II/NT-I comparison yielded 3480 (28.53%) DE mRNAs and 41 (12.42%) DE lncRNAs, and LT-II/LT-I yielded 1150 (11.33%) DE mRNAs and 13 (6.53%) DE lncRNAs (Figure 5B,C). These results demonstrate that transcripts changed substantially during stage I (NT-I and LT-I). Stage I is comparable to the plateau phase.

The upregulated and downregulated DE mRNAs and DE lncRNAs in NT and LT are shown in Figure 5D–G. The number of upregulated DE lncRNAs was not very different from the number of downregulated lncRNAs in NT and LT (Figure 5E,G). More upregulated DE mRNAs than downregulated mRNAs were identified in NT-II/NT-I and LT-II/LT-I (Figure 5D,F).

DE lncRNAs and DE mRNAs were also identified by comparing the normalized expression of transcripts at each time point under LT with the corresponding transcripts under NT (i.e., LT-I/NT-I, LT-II/NT-II, *p* < 0.05, |log2 (fold change)| ≥ 1). There were 26 DE lncRNAs in LT-I/NT-I, of which 8 were upregulated and 18 were downregulated, and 22 DE lncRNAs in LT-II/NT-II, of which 14 were upregulated and 8 were downregulated (Figure 5I). There was only one continuously changing DE lncRNA in LT-I/NT-I and LT-II/NT-II (Appendix A). There were more downregulated mRNAs than upregulated mRNAs in both LT-I/NT-I and LT-II/NT-II (Figure 5H). The proportions of continuously changing DE transcripts were different between lncRNAs and mRNAs, with more continuously changing mRNAs than lncRNAs (Appendix A).

### 2.5. Prediction of Target Genes and Enrichment Analysis of DE lncRNAs

Previous studies have shown that lncRNAs can interact with adjacent and distant protein-coding genes in cis (regulation of neighboring loci) and trans-acting (regulation of distal loci) modes [38]. In this study, adjacent protein-coding genes within 100 kb downstream or upstream of the lncRNA on the same chromosome were considered to be potential cis-regulated target genes [39]. Trans-acting targets were predicted based on the Pearson correlation coefficients (|r|> 0.9 and *p* < 0.01) between lncRNA and mRNA expression levels [40]. A total of 6140 and 5371 lncRNAs were identified to act in cis and trans modes, respectively. In LT-I/0 h, all 186 DE lncRNAs could act in cis with 3640 mRNA genes (Appendix A), and 167 DE lncRNAs could act in trans with 125,377 mRNA genes (Appendix A). In LT-II/LT-I, all 13 DE lncRNAs could act in cis with 237 mRNA genes (Appendix A), and 5 DE lncRNAs could act in trans with 256 mRNA genes (Appendix A). In LT-I/NT-I, all 26 DE lncRNAs could interact with 468 target genes in cis (Supplemental Appendix A), and 16 DE lncRNAs could interact with 766 target genes in trans (Appendix A). In LT-II/NT-II, all 22 DE lncRNAs could interact with 340 cis-targeted genes (Appendix A), and 17 DE lncRNAs could interact with 412 trans-targeted genes (Appendix A).

To analyze the functions of the identified DE lncRNAs, their target genes (Appendix A) were annotated. Among the enriched top Gene Ontology (GO) biological process (BP) terms, 2.98% of target genes of LT-I/0 h DE lncRNAs were involved in translation (Figure 6A), while 20.15% of target genes of LT-II/LT-I DE lncRNAs were involved in protein phosphorylation (Figure 6B), 4.10% of target genes of LT-I/NT-I DE lncRNAs were involved in auxin-activated signaling pathways (Figure 6C), and 4.23% of target genes of LT-II/NT-II DE lncRNAs were involved in cell wall organization (Figure 6D). There were several categories important to seed vigor, such as the auxin-activated signaling pathway, and regulation of organ growth.

### 2.6. lncRNA Acting as Potential Regulator of the Auxin-Activated Signaling Pathway

The results described above suggest that DE lncRNAs are mainly involved in stage I. GO enrichment analysis of the target genes of DE lncRNAs in LT-I/NT-I was performed to reveal the functions of lncRNAs in seed vigor under low temperature. The regulation of organ growth, auxin polar transport and the auxin-activated signaling pathway were significantly enriched (Figure 6C). Auxin can exert a pleiotropic effect on various aspects of plant growth and development [26], including cell elongation, cell division and differentiation, at the cellular level by altering the expression of numerous genes [24]. We focused on the auxin-activated signaling pathway which contained the largest number of genes in BPs. A total of 23 genes were targeted by five lncRNAs (Appendix A), which indicated that some lncRNAs may target more than one gene. A regulatory network was then constructed using Cytoscape (version 3.8.2, [41]) (Figure 7). There was a cis relationship between most lncRNAs and protein-coding genes. Only three lncRNA–mRNA pairs were found to interact in trans. Intriguingly, most of the target genes are SAUR family genes, which are clustered on chromosome 9 and targeted by lncRNA *MSTRG.182510.6*. Most of the genes were significantly upregulated in LT/0 h and NT/0 h but slightly downregulated in LT-II/LT-I and NT-II/NT-I and slightly upregulated in LT-I/NT-I and LT-II/NT-II (Appendix A). *SAUR* is one of the three major families of early auxin response genes. Different *SAUR* family genes have been found to accelerate cell elongation when overexpressed in Arabidopsis [32,34,35]. SAURs interact with PP2C.D phosphatases to prevent membrane H^+^-ATPases from being dephosphorylated, thus inducing cell wall acidification and plant growth [36]. One SAUR family member, *OsSAUR33*, was shown to regulate seed vigor via the sugar pathway during early seed germination [30]. Therefore, it was speculated that SAUR family genes may work cooperatively with lncRNAs in response to seed vigor. The expression patterns of SAUR family genes and related lncRNAs were further verified by qRT–PCR, and the results were consistent with the transcriptome sequencing data, suggesting that the transcriptome sequencing was reliable (Appendix A and Appendix A).

Subsequently, rapid amplification of cDNA ends (RACE) was performed to experimentally validate the 5′ and 3′ ends of *MSTRG.182510.6* (Figure 8 and Appendix A). Additionally, the full sequence of *MSTRG.182510.6* was amplified using primers targeting the extremities of the 3′ and 5′ end sequences, and up to 8 splice isoforms (*FULL*
*1-FULL*
*8*) were identified and sequenced (Figure 8C and Figure 9A and Appendix A). As a result, *FULL 1* is the most abundant, which is 889 bp in length and has a polyadenylated 3′ end but no 5′ cap structure (Figure 8D and Figure 9A). According to alignment, it is located in the region of 21,629,572–21,630,544 bp on chromosome 9. It is approximately 19.62 kb from the nearest SAUR family gene, *OsSAUR55* (Figure 9C). *FULL 1* was subsequently named SVR (seed vigor related lncRNA). Then, CPC was used to evaluate the coding potential of all the isoforms, and the coding scores of all the isoforms were below zero (Figure 9B). In addition, homology searches in Pfam and SMART revealed no functional domain matches. The results indicate that all eight isoforms represented noncoding RNAs. To analyze whether SVR is related to seed vigor, the spatial and temporal expression pattern of SVR was determined (Figure 9D and Appendix A). We found that SVR is predominantly expressed in the endosperm and is rarely expressed in other tissues. The tissue-specific expression pattern of SVR may indicate its relationship with seed vigor.

### 2.7. CRISPR/Cas9-Engineered Mutations in SVR Cause Delay of Germination

To investigate if SVR functions in seed vigor, a CRISPR/Cas9 construct with one *SVR* target was designed and transformed into rice Francis. The target site was designed to ve located at the longest ORF of SVR. Two CRISPR/Cas9 (CR)-SVR mutant lines were detected from the first-generation transgenic plants (T_0_). Sequencing analysis proved that the two transgenic lines were homozygous (Figure 10A). Two second-generation (T_1_) transgenic lines, named CR-SVR-8 and CR-SVR-11, were used for phenotypic investigation. A total of 50 seeds per replicate were imbibed with 18 mL of distilled water in Petri dishes (diameter 15 cm) at 25 ± 1 °C for 7 days and 15 ± 1 °C for 15 days, respectively, with three replicates for each treatment. The germinated seeds were counted every 8 h at 25 °C and every day at 15 °C. The germination rates of CR-SVR-8 and CR-SVR-11 were extremely decreased than WT under both NT and LT (Figure 10B–E). To verify the results, the seeds of the T_2_ generation were also germinated at 25 °C, showing the same germination trend as the T_1_ generation (Appendix A). From the above results, it is concluded that the lncRNA SVR seems important to seed vigor.

In summary, we identified an endosperm-specific and vigor-related lncRNA (SVR) from the adjacent region of the *SAUR* family gene cluster. At present, though the lncRNA that affects seed vigor has been verified, the molecular function analysis of this lncRNA is still under way.

## 3. Discussion

Although several lncRNAs have been identified in rice [42,43,44], there is still no research about the role of lncRNAs in rice seed vigor. Seed vigor is a complex and important agricultural trait that determines the potential for rapid, uniform emergence and the establishment of strong seedlings under a wide range of environmental conditions [2]. In this study, we performed the strand-specific whole-transcriptome sequencing of 15 rice seed samples collected at 0 h, stage I, and stage Ⅱ under NT and LT to determine the regulatory roles of lncRNAs in rice seed vigor. We found that the expression patterns and structural characteristics of lncRNAs were different from those of mRNAs. LncRNAs are more frequently expressed in a stage-specific manner than mRNAs. LncRNAs may interact with auxin-activated signaling genes to promote seed vigor under low temperature. Moreover, a tissue-specific lncRNA (SVR) adjacent to the *SAUR* family gene cluster was identified and proved to be related to seed vigor. This may provide important information for the further molecular functional analysis of lncRNAs.

### 3.1. lncRNAs Differ from mRNAs in Structure and Expression Pattern

In this research, a total of 6258 credible putative lncRNAs were identified, and the majority of the lncRNAs were lincRNAs, which was similar to the case in Chinese cabbage [45], cotton [46], and potato [47], but different from that in *Arabidopsis*, in which antisense lncRNAs were the major type [48,49]. In mammals, lincRNAs are typical stable products of RNA polymerase Ⅱand are nearly always processed by 5′-capping and 3′ poly (A) tail addition [50]. Some lincRNAs exhibit tissue-specific and stress-responsive expression patterns [51,52]. The lincRNA SVR identified in this study is predominantly expressed in the endosperm. The lncRNAs in this study possessed similar features as those identified in other studies in rice [53,54,55]. Compared to mRNAs, lncRNAs were shorter in length, with fewer exons, shorter ORF lengths, lower expression levels, and lower conservation.

LncRNAs were more often expressed in a stage-specific manner than mRNAs in this study. More lncRNAs specifically expressed at different stages of LT and NT, especially with respect to the 0 h time point. Emerging evidence shows that the expression of lncRNAs is extremely specific to tissue or cell type and environmental condition, arguing for the roles of lncRNAs in the regulation of plants [22,42,56]. It has been reported in maize and cassava under drought stress that drought-responsive lncRNAs exhibit higher tissue and development specificity than protein-coding genes [57,58]. The specific expression pattern of lncRNAs may be related to their functions in maintaining tissue identity, tissue development, differentiation, and stress responses. The stage-specific expression pattern of lncRNAs in this study indicated their unique role in seed vigor.

### 3.2. lncRNAs May Work Cooperatively with mRNAs to Regulate Seed Vigor through the Auxin-Activated Pathway

Seed germination is a complex biological process that involves three physiological phases: the rapid phase (phase I), plateau phase (phase II), and rapid water uptake phase (phase III). During the rapid phase (phase I), the absorption of water into the cells of dry seeds occurs as a physical process that is driven by the matrix potential. During the plateau phase (phase II), comprehensive metabolism and regulation of gene expression occur [59,60], followed by further rapid water uptake in phase III [61]. The overwhelming majority of DE mRNAs and DE lncRNAs identified in this study were found in NT (LT)-I/0 h, including 8719 (71.47%) DE mRNAs and 289 (87.58%) DE lncRNAs in NT-I/0 h and 8999 (88.67%) DE mRNAs and 186 (93.47%) DE lncRNAs in LT-I/0 h, respectively. In this study, stage I was comparable to the plateau phase, and transcripts changed greatly during stage I, which may reflect comprehensive gene expression.

Plants have evolved plenty of cellular, physiological, and morphological defenses to recognize various biotic and abiotic stress responses [62]. Over the past decade, RNA-seq has been used to identify many genes involved in seed vigor; however, the regulatory pathways responding to seed vigor are far from being understood. Emerging evidence indicates that lncRNAs play an important role as “biological regulators” for various development processes and biotic and abiotic stress responses in plants [63]. In this study, the discovery of lncRNAs extends our understanding of certain regulatory pathways in seed vigor under LT. Plants orchestrate several complex regulatory gene networks of C-REPEAT BINDING FACTOR (CBF)-COLD REGULATED (COR) (CBF-COR) signaling pathway under cold stress [64]. The lncRNA *SVALKA* has been reported to play a role in regulating *CBF1* expression, and the expression of *SVALKA* is increased after 4 h of cold treatment when the expression of *CBF1* is decreased [56]. RNAPⅡ read-through transcription of *SVALKA* results in a cryptic lncRNA, termed as *CBF1*, overlapping with *CBF1* on the antisense strand, and *CBF1* is suppressed by RNAPⅡ collision stemming from the *SVALKA*-as *CBF1* cascade [56]. In this study, auxin-activated signaling pathway was significantly enriched, by investigating the targets of differentially expressed (DE) lncRNAs of LT-I/NT-I. Additionally, most of the target genes were *SAUR* family members that clustered in chromosome 9. The lncRNA SVR had been speculated as the most possible functional lncRNA. The CRSPR/Cas 9 mutants of this lncRNA indicated it was related to seed vigor. This indicated that the lncRNA may be important to seed vigor. SVR is about 19 kb from the nearest *SAUR* family gene, *OsSAUR55*. It was speculated that the lncRNA may work cooperatively with those *SAUR* family genes to regulate seed vigor through the auxin pathway.

Hormones are important to seed vigor. It is widely known that abscisic acid (ABA) and gibberellic acid (GA) are key regulators of seed germination and work antagonistically. Apart from ABA and GA, studies have demonstrated that auxin is also involved in seed dormancy and germination [11]. Auxin performs a regulatory role through the transport inhibitor response 1 (T1R1)/additional F box protein (AFB)-Aux/indole-3-acetic acid (IAA)-auxin response factor (ARF) signaling pathway [65,66]. Auxin is generally considered to negatively regulate seed germination and act in an ABA-dependent manner [67]. The exogenous application of auxin enhanced the inhibition of seed germination by ABA in *Arabidopsis* [68,69] and delayed seed germination in wheat [70]. It has also been reported in soybean that exogenous auxin treatment represses soybean seed germination by enhancing ABA biosynthesis while impairing GA biogenesis [71]. Liu et al. [67] demonstrated that auxin can enhance ABA-mediated seed dormancy by recruiting ARF10/16 to maintain *ABI3* expression during seed imbibition. *SMALL AUXIN UP RNA* (*SAUR*) is one of the three families of early auxin response genes [24]. The expression pattern analysis showed that most of the targeted *SAUR* family genes were significantly upregulated in NT-I/0 h and LT-I/0 h, which suggested that *SAUR* family genes may function in the early germination stage. SAURs are involved in a wide range of cellular, physiological, and developmental processes [31]. *OsSAUR45* was proved to be involved in plant growth by negatively regulating auxin synthesis and transport [28]. *OsSAUR39* is also a negative regulator of auxin synthesis and transport in plant growth, and overexpression of *OsSAUR39* resulted in decreased levels of free IAA and decreased auxin transport [29]. Knockout of *OsSAUR33* reduced rice seed vigor, and this gene interacted with *OsSnRK1A*, affecting seed vigor through the sugar pathway [30]. In this study, most *SAURs* also respond to cold stress (Appendix A), and thus we speculated that the lncRNA SVR may work cooperatively with *SAUR* family genes to regulate seed vigor through the auxin pathway. The delay of germination in CRISPR/Cas9-engineered mutations of SVR may be caused by the level of auxin. At present, we only know the lncRNA SVR is related to seed vigor, and the molecular relationship between SVR and *SAUR* family genes is unclear. Additionally, investigations into the molecular function of this lncRNA are underway. These studies may provide useful information for the further molecular investigation of lncRNAs.

## 4. Materials and Methods

### 4.1. Seed Germination and Sampling

The cold-tolerant Japonica rice cultivar Francis was used in this study. A total of 250 seeds per replicate were imbibed with 18 mL of distilled water in Petri dishes (diameter 15 cm) at 25 ± 1 °C for 7 days (normal temperature, NT) or 15 ± 1 °C for 14 days (low temperature, LT), with three replicates for each treatment. Dry seeds were designated as 0 h, and the treatments were designated as NT (LT)-I and NT (LT)-II. The time points at which the embryos of 80% of the seeds just broke out of the glumes and at which the embryo length of 80% of the seeds reached 1.0 mm were designated as NT (LT)-I and NT (LT)-II, respectively. Three replicates were established for each treatment, and 30 seeds per replicate were collected at the 0 h time point, NT (LT)-I and NT (LT)-Ⅱ, for subsequent sequencing. A total of 15 samples were quickly placed in liquid nitrogen and immediately stored at −80 °C for sequencing.

### 4.2. RNA Extraction, Library Construction, and Strand-Specific Sequencing

Total RNA was isolated from the seeds at different stages under NT and LT using an EASYspin Plus Complex Plant RNA Kit (Aidelai, Beijing, China) according to the manufacturer’s instructions. To remove contaminating DNA, DNase I (Promega, Madison, WI, USA) was added to the extracted RNA. Before library construction, the quality and integrity of the RNAs were verified using a Nanodrop 2000 instrument and Agilent Bioanalyzer 2100 System (Agilent Technologies, Santa Clara, CA, USA).

For library construction, 1.5 µg of total RNA from each sample was used to remove rRNA with the Ribo-Zero rRNA Removal Kit (Epicentre, Madison, WI, USA). Sequencing libraries were constructed using the NEBNextR UltraTM Directional RNA Library Prep Kit for Illumina (NEB, Ipswich, MA, USA) following the manufacturer’s instructions. To select insert fragments of 150~200 bp in length, the library fragments were purified with AMPure XP beads (Beckman Coulter, Beverly, Australia; Brea, CA, USA). Then, the size-selected and adaptor-ligated cDNA was treated with 3.0 μL of USER Enzyme (NEB, USA) for 15 min, and PCR was performed with Phusion High-Fidelity DNA polymerase, Universal PCR primers, and Index Primer. The PCR products were purified with the AMPure XP system, and the library quality was assessed on an Agilent Bioanalyzer 2100. Finally, the purified libraries were sequenced on an Illumina NovaSeq 6000 platform at Biomarker Biotechnology Co., Ltd. (Beijing, China) with a 100 bp paired end. Three biological replicates were performed for each sample.

### 4.3. Identification of lncRNAs

The Japonica reference genome IRGSP-1.0 (https://rapdb.dna.affrc.go.jp/download/history/irgsp1_2019-06-26.html (accessed on 15 July 2019)) was downloaded. Clean reads were obtained from the raw sequencing data by removing reads containing adapters, poly-N, and low-quality reads. Then, the clean reads of each sample were mapped onto the Japonica reference genome using the HISAT2 program (version 2.0.4, [37]). After alignment, the transcripts were assembled and merged to generate final transcripts using String Tie (version 1.3.1). Then, the GffCompare program [72] was applied to annotate the assembled transcripts. The unknown transcripts were screened for putative lncRNAs. The fragments per kilobase of transcript per million mapped reads (FPKM) values for each transcript were calculated using String Tie (version 1.3.1). Transcripts with FPKM scores < 0.1, lengths shorter than 200 nucleotides, or fewer than two exons were discarded. Then, the Coding Potential Calculator (CPC, score < 0), Coding Non-Coding Index (CNCI, score < 0), Coding Potential Assessment Tool (CPAT, score < 0) and Pfam-scan (version 1.3, E-value < 0.01) were combined to screen the candidate lncRNAs. The lncRNAs were classified as lincRNAs, intronic lncRNAs, antisense lncRNAs, or sense lncRNAs by Cuffcompare (version 2.1.1).

### 4.4. Prediction of Target Genes

The targets of lncRNAs were predicted through two independent algorithms: cis or trans [73]. Cis-acting lncRNAs act on neighboring target genes; thus, genes within a 100 kb window upstream or downstream of lncRNAs were defined as potential cis target genes [39]. The algorithm search for trans target genes was based on the correlation of expression levels by Pearson correlations coefficient. When the absolute value of the correlation coefficient was >0.90 and q < 0.01, a gene was predicted to be a trans target of a lncRNA [40].

### 4.5. Analysis of Differential Expression Patterns and Enrichment Analysis

Differentially expressed lncRNAs (DE lncRNAs) and differentially expressed mRNAs (DE mRNAs) between NT and LT at different stages of germination were determined using the DESeq R package (1.10.1) with |log2 (fold change)| ≥1 and an FDR (false discovery rate) threshold of 5% [74]. To annotate the putative target genes of lncRNAs, the following databases were used: Nr (NCBI nonredundant protein sequences), Pfam (Protein family), KOG/COG (Clusters of Orthologous Groups of proteins), Swiss-Prot (a manually annotated and reviewed protein sequence database), KEGG (Kyoto Encyclopedia of Genes and Genomes), and GO (Gene Ontology). The topGO R package was used for the Gene Ontology (GO) enrichment analysis, and KEGG pathway enrichment analysis was performed using KOBAS software [75].

### 4.6. Network Construction

The target genes of DE lncRNAs from LT-I/NT-I related to the auxin-activated signaling pathway were searched throughout the whole dataset. All 23 protein-coding genes were used as putative targets to study their interactions with DE lncRNAs. Then, the lncRNA–mRNA network was visualized using Cytoscape software (version 3.8.2, [41]).

### 4.7. qRT–PCR

To validate the expression of lncRNAs and mRNAs in different samples, qRT–PCR was performed. The EASYspin Plus Complex Plant RNA Kit (Aidelai, Beijing, China) was used to extract RNA from samples. Then, approximately 1.0 µg of total RNA was reverse transcribed to cDNA with random primers and oligo (dT) primers. qRT–PCR was performed on an ABI StepOne Real-Time PCR system (Applied Biosystems, Waltham, MA, USA) using AceQ^®^ qPCR SYBR Green Master Mix (Vazyme, Nanjing, China) for three replicates per sample. *OsActin* (LOC_Os03g50885) was the reference gene for mRNAs and lncRNAs. The ^ΔΔ^CT method was used to normalize the relative expression levels of mRNAs and lncRNAs [76]. The endosperm, radicle and germ were sampled from the germinated seeds imbibed in five days in dark under 30 °C for tissue-specific expression analysis. The 3-leaf-stage leaves and 3-leaf-stage stems were sampled from the Francis plants at a paddy field with conventional cultivation management. The flag leaf, leaf sheath, young panicles and nodes were sampled from the Francis plants during the flowering stage at the paddy field with conventional cultivation management.

### 4.8. RACE Analysis

The RNA samples were extracted from germinated seeds of LT-I and stored at −80 °C. The 3′ RACE analysis was completed using SMARTER RACE 5′/3′ Kit (Takara, Japan) following the manufacturer’s instructions, the same as 5′ RACE. The 3′ end products were obtained by strand-specific amplification using 10× Universal Primer A Mix (UPM) and 3′ Nested Gene Specific Primer (NGSP), and the 5′ end products were obtained using 10× UPM and 5′ NGSP (Appendix A). Subsequently, the 5′ and 3′ RACE products were characterized by sequencing. To obtain the full-length cDNA sequence, the cDNA was subjected to strand-specific amplification with the 5′-end primer and the 3′-end primer and then the products were characterized by sequencing (Appendix A).

### 4.9. Construction of CRISPR/Cas9 Vector and Transgenic Plants

For CRISPR/Cas9 vector construction, the tRNA-target site-gRNA gene-editing system was used. SgRNAs were designed according to CRSPR-P web tool [77]. One sgRNA was designed, namely sgRNA1. The target site was located at the longest open reading frame (ORF). The pRGEB32 was used as the construction vector [78]. The constructed vector was then transformed into *Agrobacterium* strain by electroporation. Francis rice was used as the material, the seeds were sterilized and placed on callus induction medium in dark at 25 °C for about 7 days, then the healthy callus tissues were transferred to subculture medium and co-cultured with *Agrobacterium*. The genetic transformation was performed using previously reported method [79].

## Figures and Tables

**Figure 1 ijms-23-09472-f001:**
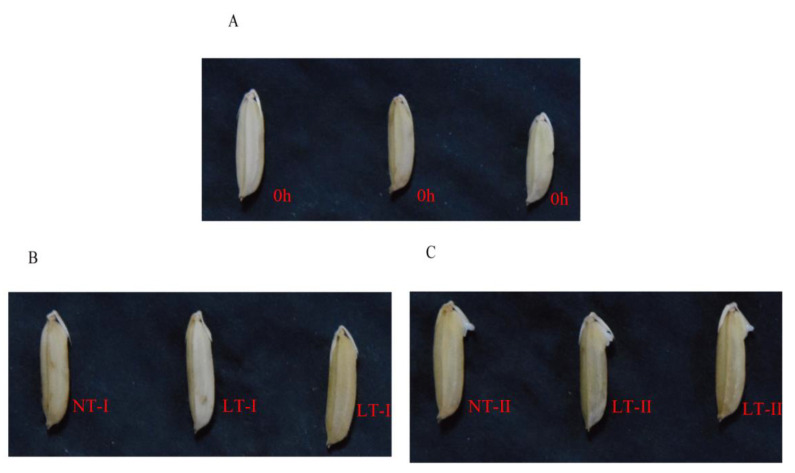
Dynamic changes in seed germination of cultivar Francis under NT (25 °C) and LT (15 °C) conditions. (**A**) 0 h: untreated dry seeds. (**B**) NT-I or LT-I: seeds germinated at 25 °C or 15 °C; the embryos had just broken out of the glumes. (**C**) NT-II or LT-II: seeds germinated at 25 °C or 15 °C; embryo length was ≥1.0 mm.

**Figure 2 ijms-23-09472-f002:**
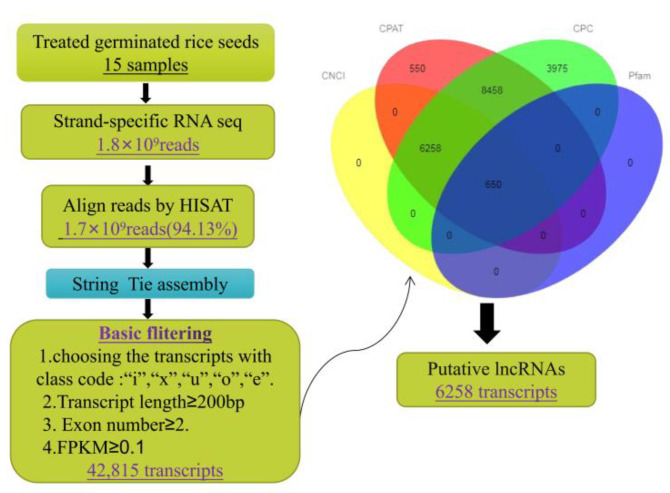
Pipeline of the prediction of lncRNAs in Francis. CNCI, coding non-coding index; CPC, coding potential calculator; CPAT, Coding Potential Assessment Tool; Pfam, a database of protein families.

**Figure 3 ijms-23-09472-f003:**
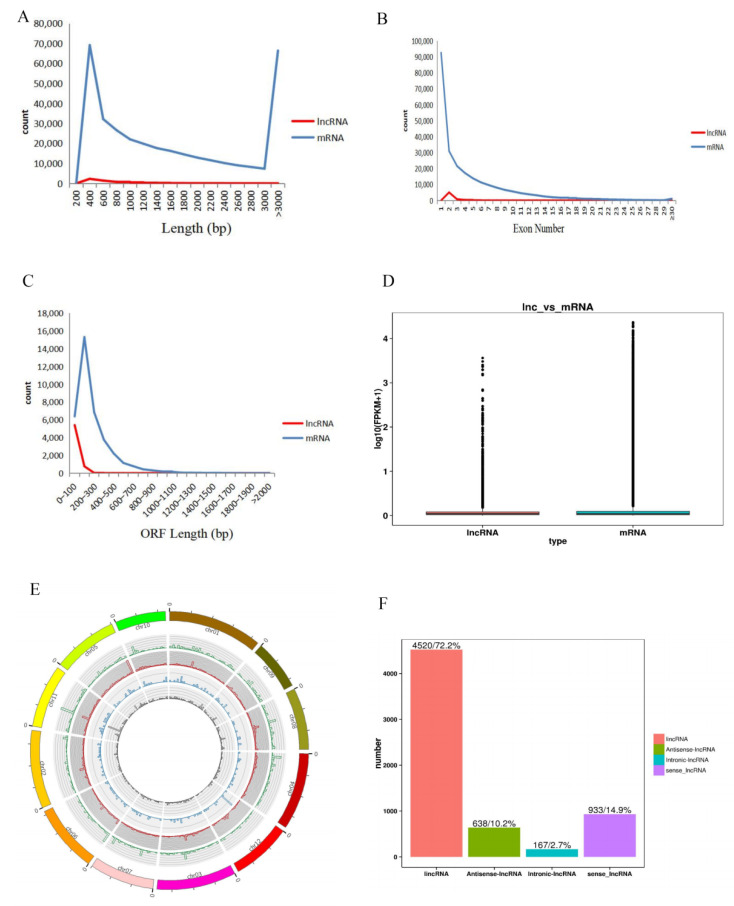
Differences in structure and expression between lncRNAs and mRNAs. (**A**) Length distributions of lncRNAs and mRNAs. (**B**) Distribution of predicted exon numbers in lncRNAs and mRNAs. (**C**) Distribution of ORF (open reading frame) length. (**D**) Overall expression levels of lncRNAs and mRNAs in Francis. (**E**) The distribution of lncRNAs among 12 chromosomes. The four concentric rings correspond to the expression levels of sense lncRNAs, lincRNAs, intronic lncRNAs, and antisense lncRNAs, from outer to inner. (**F**) The proportions of different types of lncRNAs.

**Figure 4 ijms-23-09472-f004:**
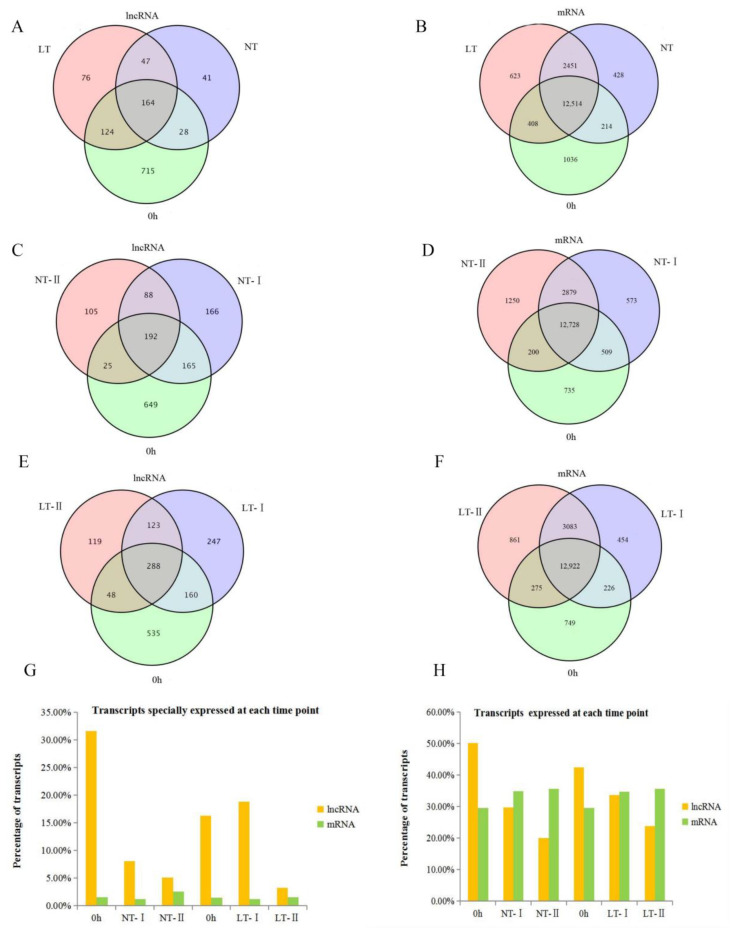
The characteristics of the expression of lncRNAs and mRNAs during seed germination under NT and LT. (**A**) Venn diagram of expressed lncRNAs under NT (NT-I, NT-II), LT (LT-I, LT-II) and 0 h. (**B**) Venn diagram of expressed mRNAs under NT (NT-I, NT-II), LT (LT-I, LT-II) and 0 h. (**C**) Venn diagram of lncRNAs expressed under NT. (**D**) Venn diagram of mRNAs expressed under NT. (**E**) Venn diagram of lncRNAs expressed under LT. (**F**) Venn diagram of mRNAs expressed under LT. (**G**) Percentage of specifically expressed lncRNAs and mRNAs at different stages of NT and LT. (**H**) Percentage of totally expressed lncRNAs and mRNAs at different stages of NT and LT.

**Figure 5 ijms-23-09472-f005:**
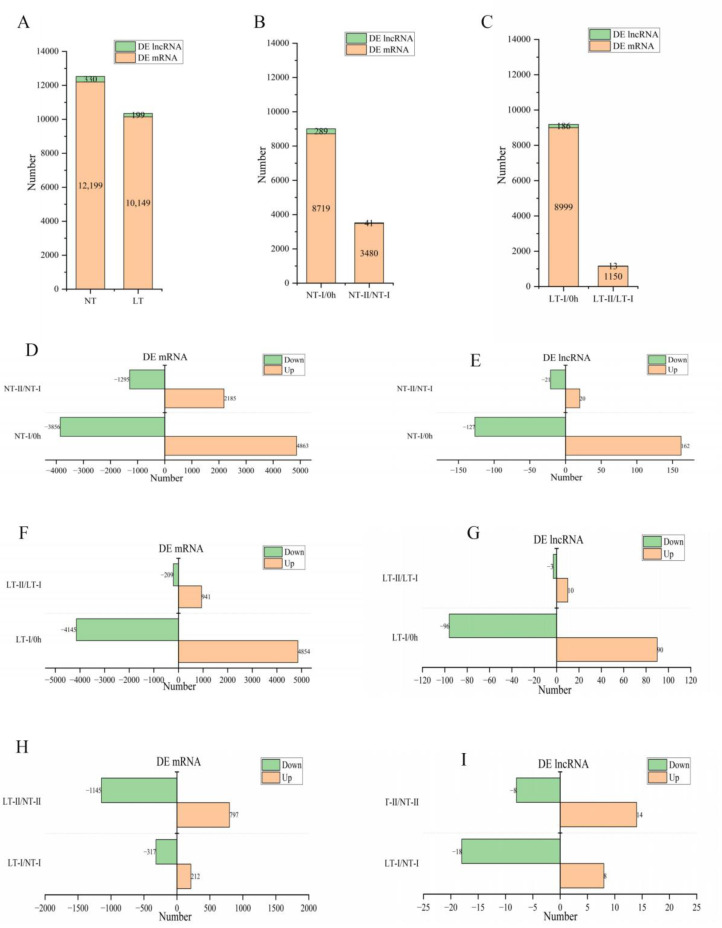
DE lncRNAs and DE mRNAs at different stages of NT and LT. (**A**) Number of total DE transcripts under NT and LT treatments. (**B**) Number of DE lncRNAs and DE mRNAs under NT treatments. (**C**) Number of DE lncRNAs and DE mRNAs under LT treatments. (**D**) Number of upregulated and downregulated DE mRNAs under NT treatments. (**E**) Number of upregulated and downregulated DE lncRNAs under NT treatments. (**F**) Number of upregulated and downregulated DE mRNAs under LT treatments. (**G**) Number of upregulated and downregulated DE lncRNAs under LT treatments. (**H**) Upregulated and downregulated DE mRNAs in LT-I/NT-I and LT-II/NT-II. (**I**), Upregulated and downregulated DE lncRNAs in LT-I/NT-I and LT-II/NT-II.

**Figure 6 ijms-23-09472-f006:**
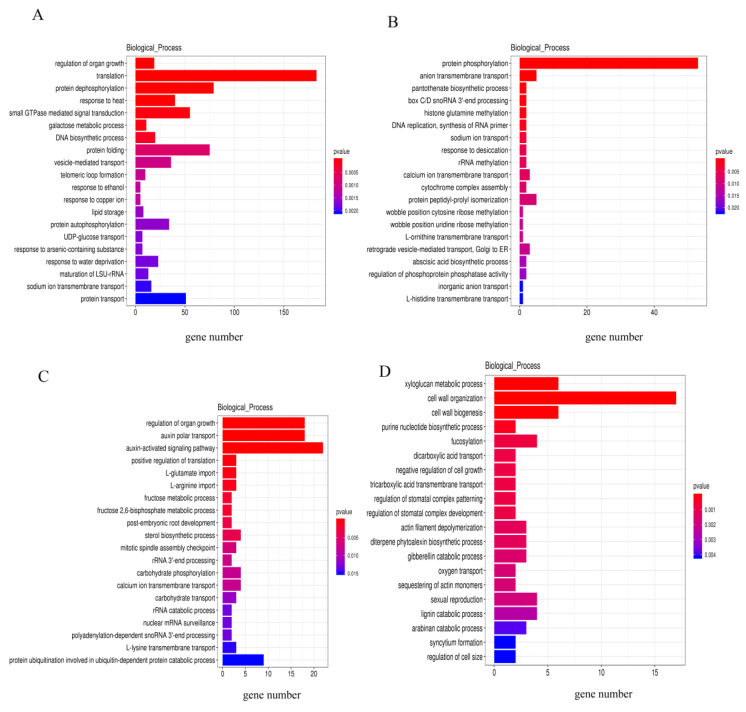
GO enrichment analysis of genes targeted by the DE lncRNAs. (**A**) Biological process terms enriched among target genes of LT-I/0 h DE lncRNAs. (**B**) Biological process terms enriched among target genes of LT-II/LT-I DE lncRNAs. (**C**) Biological process terms enriched among target genes of LT-I/NT-I DE lncRNAs. (**D**) Biological process terms enriched among target genes of LT-II/NT-II DE lncRNAs.

**Figure 7 ijms-23-09472-f007:**
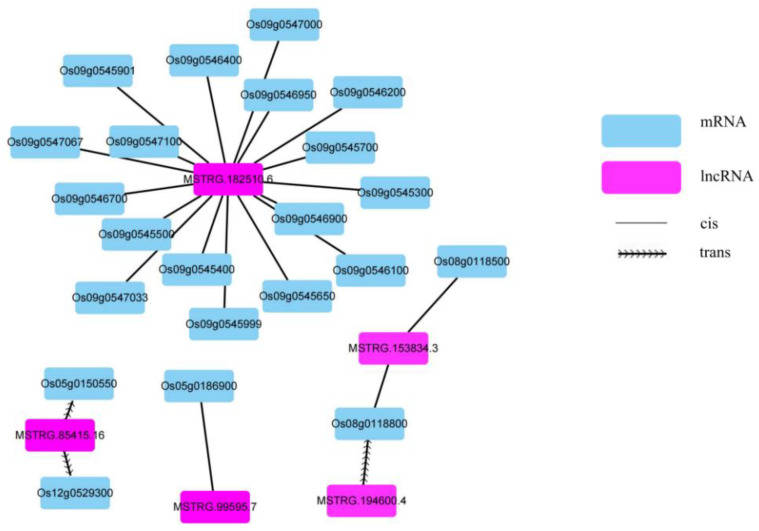
The regulatory network connecting lncRNAs and auxin-activated signaling pathway mRNAs. The purple diamonds represent lncRNAs, and the blue diamonds represent mRNAs. The wavy lines represent *trans* relationships, and the straight lines represent cis relationships.

**Figure 8 ijms-23-09472-f008:**
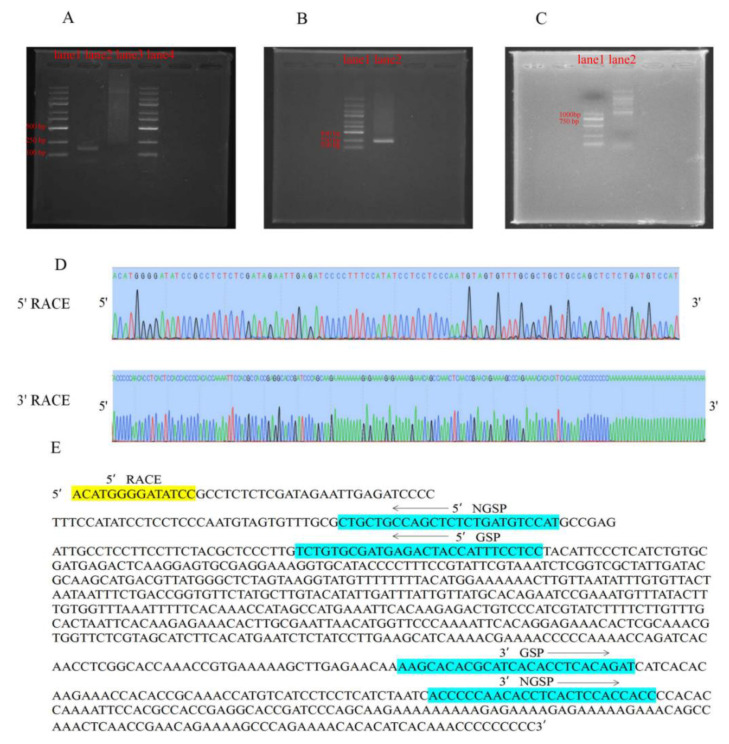
The results of 3′ RACE and 5′ RACE of SVR. (**A**) Strand-specific amplification of SVR using 3′ RACE cDNA fragments. 3′ NGSP-forward primer (presented in (**E**)) with UPM was used to obtain 3′ end products, which are presented in lane 2. Lane 1: DNA marker M5000. Lane 3: positive control. (**B**) Strand-specific amplification of SVR using 5′ RACE cDNA fragments. 5′ NGSP-reverse primer (presented in (**E**)) with UPM was used to obtain 5′ end products, which are presented in lane 2. Lane 1: DNA marker M5000. (**C**) Strand-specific amplification of full-length cDNA of SVR with 5′-end primers and 3′-end primers, which are presented in lane 2. Lane 1: DNA marker M2000. (**D**) Peak diagram of 3′ RACE products and 5′ RACE products. The 3′ RACE products and 5′ RACE products of SVR were characterized by sequencing. (**E**) Sequence of the predicted SVR. The 5′ RACE sequences are indicated in yellow, and 3′ RACE sequences are not indicated for the same to RNA-Seq sequences. The 5′ NGSP (nested gene specific primer) and GSP (gene specific primer) are indicated in blue, the 3′ NGSP and GSP are also indicated in blue.

**Figure 9 ijms-23-09472-f009:**
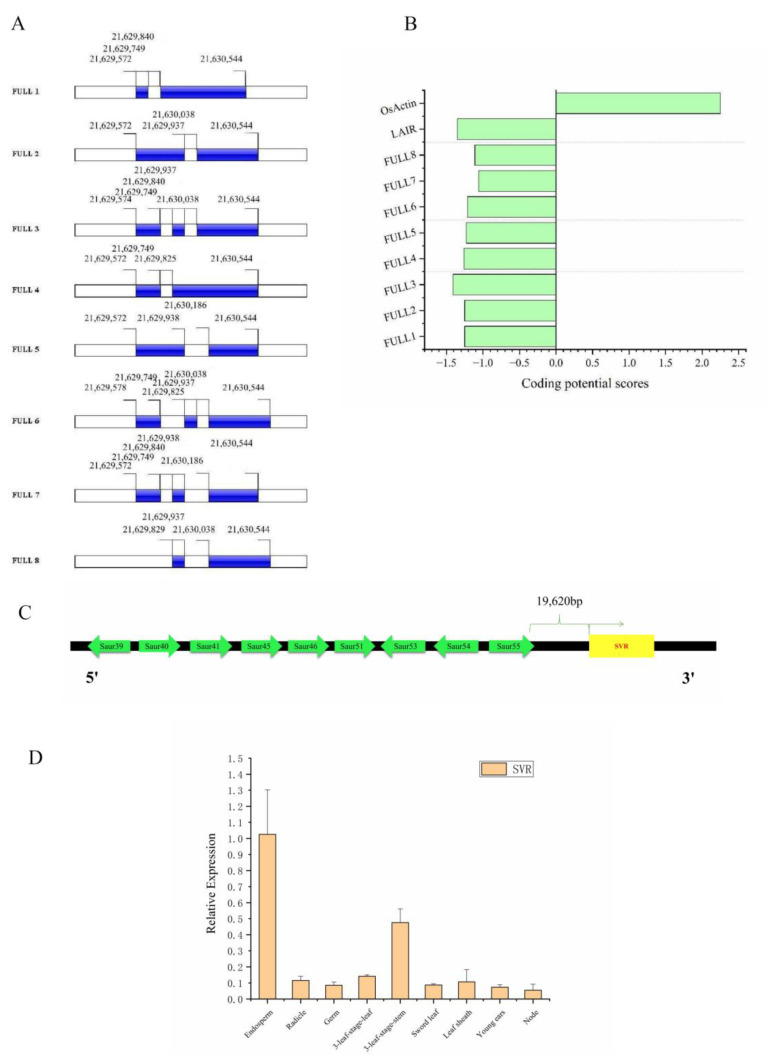
The genomic location, coding potential analysis, alternatively spliced isoforms and the spatial and temporal expression pattern of SVR. (**A**) The genomic locations of alternatively spliced SVR isoforms. *FULL 1* represents SVR. (**B**) The coding potential scores of the isoforms calculated by CPC. Transcripts with scores beyond -1 and 1 are marked as noncoding or coding in this CPC classification. *LAIR* [13] and *OsActin* represent noncoding and coding examples, respectively. (**C**) Schematic diagram of the SAUR family gene cluster and SVR. SVR is 19.62 kb from the nearest SAUR family gene, *OsSAUR55*. (**D**) The spatial and temporal expression pattern of SVR.

**Figure 10 ijms-23-09472-f010:**
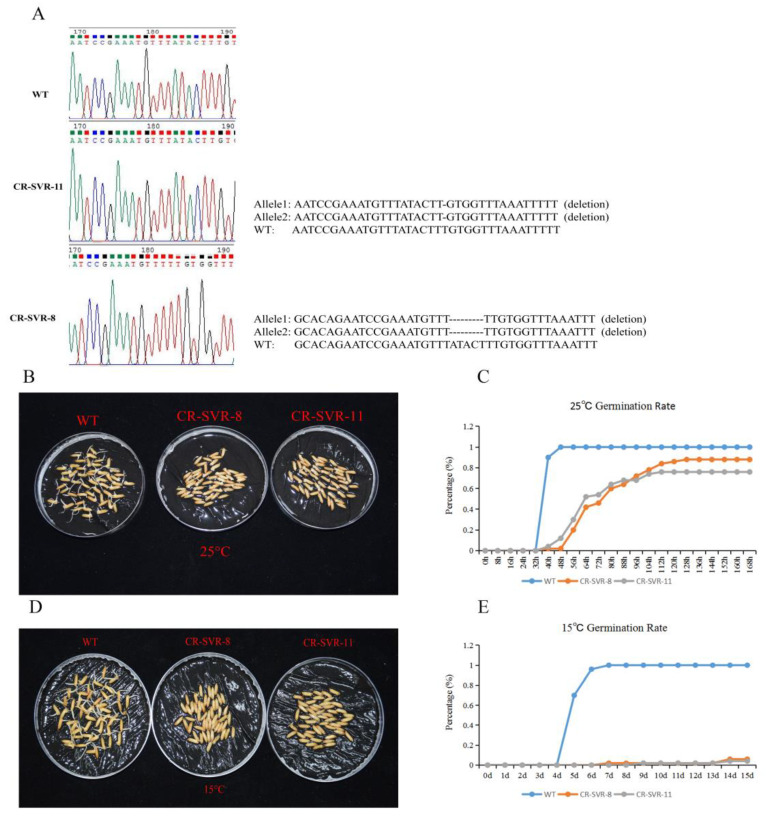
CRISPR/Cas9-engineered mutations in SVR cause delay of germination in the T_1_ generation. (**A**) Genotype of CR-SVR-8 and CR-SVR-11 in the T_1_ generation. (**B**) Germination status of two independent homozygous mutants CR-SVR-8 and CR-SVR-11 were compared with WT in 64 h imbibition at 25 °C. (**C**) Germination rate of CR-SVR-8 and CR-SVR-11 at 25 °C. (**D**) Germination status of two independent homozygous mutants CR-SVR-8 and CR-SVR-11 were compared with WT in 15d imbibition at 15 °C. (**E**) Germination rate of CR-SVR-8 and CR-SVR-11 at 15 °C.

## Data Availability

The sequencing data have been deposited in the Sequence Read Archive (SRA) at the National Center for Biotechnology Information (NCBI) under the accession number PRJNA793516. Reviewer link: https://dataview.ncbi.nlm.nih.gov/object/PRJNA793516?reviewer=1247a638jpv3qpev9qhsgr8dmc (accessed on 31 May 2022.)

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
