# Peer review of "A Long Noncoding RNA Derived from lncRNA–mRNA Networks Modulates Seed Vigor"

_ijms, 2022, doi:10.3390/ijms23169472_

Round 1
Reviewer 1 Report
The MS entitle; A Long Noncoding RNA Derived from lncRNA–mRNA Networks Modulates Seed Vigor, I well designed, and excellent study. this study can provide an excellent background for abiotic stress tolerance.
The following question must be addressed;
· Why seed germinations;
· Author must provide more supported data such as philological and biochemical data that can define, yes that’s can influence seed germination.
· These is no any evidence that’s can support that SUAR are involved in seed germination! Explant
· Why no any evidence regarding plant growth, leaf, seed size and plant growth and development? If its effect seed germination, they must influence plant growth.
· Figure 10. CRISPR/Cas9-engineered mutations in SVR……….why only 15 and 25C, provide data that’s mutant and WT grow normally. Also provide and extra experiment that’s can grow under 38-40C, to make sure that’s can influence seed germination.
· Why mutant seed can’t grow under both condition.
· Plant physiological analysis are needed. add SAUR family genes expression.
Author Response
Dear Reviewer,
We gratefully appreciate you for reading and giving valuable suggestions, which has significantly improved the presentation of our manuscript. Thank you very much! We have studied the comments carefully and made revision, which has resulted in a paper that is clearer, more compelling, and broader. The revised manuscript was attached, we are grateful for your kind consideration. Below the comments are responded point by point and the revisions are indicated in the new versions.
- Why seed germinations;Author must provide more supported data such as philological and biochemical data that can define, yes that’s can influence seed germination.
Response:
Thanks for your suggestions. The topic of this article is about seed germination. Seed germination is a big issue, there are too many factors influence seed germination. Since SAUR is the largest family of early auxin-responsive genes, we focused on the effect of auxin on germination, which has been addressed in the discussion section. Due to time constraints, we cannot provide biochemical data to define. We will perform biochemical analysis in our next research, thank you so much for your suggestions! Whether the lncRNA and its potential target genes, the adjacent SAUR family genes, affect other traits or not will be studied next.
- These is no any evidence that’s can support that SUAR are involved in seed germination! Explant
Response:
Thank you for pointing out this issue. Most of the SAUR genes clustered in Chromosome 9 are proved to response to cold stress which was attached in Supplemental Figure 5, and it may suggest the SAUR family genes are related to cold stress too. The CRSPR/CAS9 mutation of the SAUR family is also on the way. Besides, OsSAUR33 has been reported to affect seed vigor.
- Why no any evidence regarding plant growth, leaf, seed size and plant growth and development? If its effect seed germination, they must influence plant growth.
Response:
Thank you for pointing out this issue. Through our observation on T1 generation, there are no obvious effects on plant growth, leaf, seed size and plant growth and development under normal temperature. Since it affects seed germination, the growth of the knock-out seedlings really retarded slightly under normal temperature. The general plant growth has not been affected significantly at mature stage. The data has not been provided in this manuscript. As for low temperature, we have not tested for the limit of experimental conditions and transgenic seeds. Therefore, we focused on seed vigor especially under low temperature in this paper.
- Figure 10. CRISPR/Cas9-engineered mutations in SVR……….why only 15 and 25C, provide data that’s mutant and WT grow normally. Also provide and extra experiment that’s can grow under 38-40C, to make sure that’s can influence seed germination.
Response:
Thanks for the suggestion. The SVR was identified from cold stress, and we mainly focused on germination under cold stress, the germinated seeds for RNA-Seq was also sampled at normal temperature 25℃ and low temperature 15℃, so we just performed experiments on 25℃ and 15℃.We think it is not necessary to perform experiments under 38-40℃ in this paper. However, we will carry out the seed germination tests under high temperature in future. Thanks very much for your suggestions again.
- Why mutant seed can’tgrow under both condition.
Response:
Thank you for pointing out this issue. At present, we cannot figure out the mechanism why mutant seeds cannot germinate like the WT under both condition. We can just conclude that it may be important to seed vigor. Through the data of RNA-Seq , the SVR predominantly expressed in LT-Ⅰ(Supplementary Figure 2). And we speculated that it may be caused by auxin level, due to time and transgenic seeds constraint, we cannot provide the results at present. We have presented the germination results of the T2 generation seeds under 25℃, which was attached in Supplementary Figure 4. The results showed the same germination trend as the T1 generation seeds. We hope this can help verify the results.
- Plant physiological analysis are needed. add SAUR family genes expression.
Response:
Thanks for your suggestions. We are sorry about the physiological analysis, we cannot present the results for the time being, due to time constraints. Besides, we did not obtain enough seeds to do it. However, we really appreciate your suggestions and it is very good for understanding the difference in seed germination. We will perform the physiological analysis in following research. The expression pattern of SAUR family genes responding to cold stress has been attached in Supplementary Figure 5. According to the results, most of the SAUR family genes that cluster in Chromosome 9 has responded to cold stress, which may suggest the SAUR family genes are related to cold stress too.
We would like to express our great appreciation to you for valuable comments on our manuscript. Looking forward to hearing from you.
Best wishes,
Sincerely,
Professor Hui Wang
Reviewer 2 Report
The authors identified genome-wide rice lncRNAs involved in the regulation of seed vigor. They analyzed the trascriptome using the samples of different temperatures and germination stages. Moreover, they analyzed the putative targets of SVRs by correlation analysis. Overall, the authors provided large-scale information about the lncRNA regulators involved in rice seed vigor. And it will be very helpful to other researchers. I endorse the manuscript for publication and have several comments for improving the manuscript.
1. In Figure 1, the author have to clarify each picture. In Figure 1B, which is NT-1 or LT-1? In Figure 1C, which is NT-II or LT-II? d
2. In Figure 6, there was no legend for x-axis.
3. In Figure 8, mark the lane numbers and marker size in Figure A, B, and C.
4. In Figure 9D, there was no legend for y-axis.
5. In Figure 10, the authors performed the germination analysis using WT and two transgenic alles (T1 generation). However, somaclonal variation during transformation generally affects the seed vigor and decrease the seed germination efficiency, especially, in early transgenic generation. Therefore, the authors have to test the germination rate using transgenic line which has not been editted as a control (not WT).
Author Response
Dear Reviewer,
We gratefully appreciate you for reading and giving valuable suggestions, which has significantly improved the presentation of our manuscript.
We have studied the comments carefully and made revision, which has resulted in a paper that is clearer, more compelling, and broader. The revised manuscript was attached, we are grateful for your kind consideration. Below the comments are responded point by point and the revisions are indicated in the new versions.
- In Figure 1, the author have to clarify each picture. In Figure 1B, which is NT-1 or LT-1? In Figure 1C, which is NT-II or LT-II? D
Response:
Thank you for pointing out this issue. We have clarified each picture in Figure1.
- In Figure 6, there was no legend for x-axis.
Response:
Thanks for reminding. We have added the legend of x-axis.
- In Figure 8, mark the lane numbers and marker size in Figure A, B, and C.
Response:
Thanks for the advice. We have marked the lane numbers and marker size in Figure 8A and 8B.
- In Figure 9D, there was no legend for y-axis.
Response:
Thanks for pointing out this issue. We have added the legend of y-axis.
- In Figure 10, the authors performed the germination analysis using WT and two transgenic alleles (T1generation). However, somaclonal variation during transformation generally affects the seed vigor and decrease the seed germination efficiency, especially, in early transgenic generation. Therefore, the authors have to test the germination rate using transgenic line which has not been edited as a control (not WT).
Response:
Thank you so much for your valuable suggestions. We did not keep the transgenic line which has not been edited. We will take your advice in the future transgenic plants creating. To verify the results, the seeds of the T2 generation were also germinated under 25℃ (Supplemental Figure 4), it was shown the same germination trend as the T1 generation. So we can exclude the effect of somaclonal variation.
We would like to express our great appreciation to you for valuable comments on our manuscript. Thank you so much! Looking forward to hearing from you.
Best wishes,
Sincerely,
Professor Hui Wang